# Surface Passivation Using N-Type Organic Semiconductor by One-Step Method in Two-Dimensional Perovskite Solar Cells

Helong Wang [1,2], Guanchen Liu [3], Chongyang Xu [4], Fanming Zeng [1,*], Xiaoyin Xie [3,5,*] and Sheng Wu [4,*]

1   School of Materials Science and Engineering, Changchun University of Science and Technology, Changchun 130022, China; whljls@126.com
2   School of Mechanical Engineering, Jilin Railway Technology College, Jilin 132200, China
3   School of Chemistry and Chemical Technology, Hubei Polytechnic University, Huangshi 435003, China; liuguanchen1123@163.com
4   Yantai Research Institute and Graduate School of HEU, Harbin Engineering University, Harbin 150001, China; xcy666@163.com
5   Department of Physics, Gachon University, Gyeonggi 13120, Korea
*   Correspondence: zengfm@126.com (F.Z.); xyxie@hbpu.edu.cn (X.X.); wusheng@hrbeu.edu.cn (S.W.)

**Abstract:** Surface passivation, which has been intensively studied recently, is essential for the perovskite solar cells (PSCs), due to the intrinsic defects in perovskite crystal. A series of chemical or physical methods have been published for passivating the defects of perovskites, which effectively suppressed the charge recombination and enhanced the photovoltaic performance. In this study, the n-type semiconductor of [6,6]-phenyl-$C_{61}$-butyric acid methyl ester (PCBM) is dissolved in chlorobenzene (CB) for the surface passivation during the spin-coating process for depositing the two-dimensional (2D) perovskite film. This approach simplifies the fabrication process of 2D PSCs and benefits the film quality. As a result, the defects of perovskite film are effectively passivated by this method. A better perovskite/PCBM heterojunction is generated, exhibiting an increased film coverage and improved film morphology of PCBM. It is found that this technology results in an improved electron transporting performance as well as suppressed charge recombination for electron transport layer. As a result, PSCs based on the one-step formed perovskite/PCBM heterojunctions exhibit the optimized power conversion efficiency of 15.69% which is about 37% higher than that of regular perovskite devices. The device environmental stability is also enhanced due to the quality improved electron transport layer.

**Keywords:** perovskite solar cells; electron transport layer; 2D perovskite; anti-solvent

## 1. Introduction

Organo-inorganic hybrid perovskite solar cells (PSCs) have grabbed considerable attention due to their excellent characters, such as strong light absorption, long carrier diffusion length, high charge carrier mobility, and cost-effective production [1–5]. Within one decade of development, the power conversion efficiency (PCE) of PSCs has been over 25% [6], reaching a comparable level to those of their commercialized counterparts, e.g., cadmium telluride (CdTe), and copper indium gallium selenide (CIGS) solar cells [7]. Moreover, because of the nature of their mechanical flexibility, PSCs are compatible with the continuous roll-to-roll production, a kind of batch production, exhibiting the potential for a wider range of practical application [8–12]. However, due to the ionic bond and crystal structure, perovskite devices suffer from the poor environmental stability and moisture resistance which could obviously accelerate the degradation rate of perovskites [13–17]. A lot of effort has been made to solve such defects, including element doping, solvent engineering, and interfacial modification [3,18–21]. These methods yield the perovskite film with a better quality by modifying the crystal structure, optimizing the crystal configuration, and improving the interfacial morphology, which means decreasing the defects

and increasing the regularity. Many methods have been studied in three-dimensional PSCs, which are also meaningful for the two-dimensional (2D) PSCs [22–26]. Due to the special character of 2D perovskites, passivating the defects in 2D perovskite may be more significant. Generally, the 2D perovskites can be presented with the molecular formula of $A_2B_{n-1}M_nX_{3n+1}$, where M stands for metal ion, e.g., $Pb^{2+}$ or $Sn^{2+}$, X presents halide ion, A stands for spacer cation, such as $CH_3(CH_2)_3NH_3^+$ ($BA^+$) or $C_6H_5(CH_2)_2NH_3^+$ ($PEA^+$), B is $CH_3NH_3^+$ ($MA^+$), $HC(NH_2)_2^+$ ($FA^+$), or $Cs^+$ [27]. N stands for the number of metal halide monolayer sheets. When increasing the n value from 1 to ∞, the perovskite structure gradually turns from pure 2D phases into 3D perovskites [28]. The 2D perovskite sheets are surrounded with the organic spacers which are hydrophobic and could provide robust moisture resistance [29]. That is one of the crucial reasons why 2D perovskite devices usually exhibit excellent environmental stability. However, these spacer molecules can effectively suppress the charge transfer when n values are low (i.e., n < 5) [30]. In addition, it demanded more optimization for depositing the functional layer on top of the 2D perovskite due to its hydrophobic character. Therefore, 2D PSCs usually exhibit a lower PCE compared with their 3D counterparts.

Many efforts have been made to elevate photovoltaic performance of 2D perovskite devices, including optimizing crystal orientation and crystallographic texture of 2D perovskites [27,29]. H. Chen's group obtained the vertically orientated 2D perovskites with high crystal quality by adding an appropriate amount of $NH_4SCN$ into perovskite precursor as the additive, achieving a PCE of 11.01% [31]. A. D. Mohite et al. obtained 2D light absorber with vertical orientation using a hot casting approach, yielding an optimized PCE of 12.5% with well suppressed photocurrent hysteresis [29]. Their developed 2D PSCs presented obviously enhanced environmental stability compared with the 3D counterparts. Compared with the research directions about developing the 2D perovskite crystal quality, optimizing the charge transport layers (CTLs), i.e., electron transport layer (ETLs) and hole transport layers (HTLs), seems less popular. However, in the field of 3D PSCs, developing the CTLs are quite meaningful for boosting the photovoltaic performance of devices, e.g., the open circuit voltage ($V_{oc}$) and short circuit current ($J_{sc}$) can be obviously improved by optimizing the ETLs [32–34]. C. Xu developed the ETL of 3D planar inverted PSCs by replacing conventional [6,6]-phenyl-C61-butyric acid methyl ester (PCBM) with the fullerene mixture of C60/C70 and achieved a PCE improved from 15.2 to 16.9%[19]. J. Huang's group obtained the structural order of PCBM layer using the solvent annealing process and the PCE of perovskite sample was improved from 17.1 to 19.4% [33]. They argued that the quality improved ETL had a significant impact on the performance of PSCs and the $V_{oc}$ was effectively enhanced. Hence, developing the ETL is quite meaningful and necessary for improving the performance of 2D PSCs.

In this work, we report a facile approach to improve the photovoltaic performance and environmental stability of inverted 2D PSCs by passivating the 2D perovskite with the PCBM solution. Regularly, the PCBM is dissolved into chlorobenzene (CB) to make the PCBM solution which is dropped on the formed perovskite film to deposit the ETL. However, we applied the one-step method (OSM) by dropping the PCBM solution as the anti-solvent during spinning the perovskite. By this way, the defects of perovskite film were effectively passivated and a better film coverage of PCBM was obtained. As a result, the 2D perovskite devices based on one-step method (OSM) exhibit a PCE of 12.8%, which is obviously elevated compared with that of control group (8.0%). At the same time, the long-term stability is also evidently enhanced. These results provide more aspects about developing the performance of 2D PSCs.

## 2. Materials and Methods

### 2.1. Chemicals and Reagents

Butylamine hydroiodide (BAI) and lead iodide ($PbI_2$) were bought from Tokyo Chemical Industry Co., Ltd. (Tokyo, Japan). *N,N*-dimethylformamide (DMF) and CB were obtained from Sigma-Aldrich (Schnelldorf, Germany), while methylammonium io-

dide (MAI) and 2,9-dimethyl-4,7-diphenyl-1,10-phenanthroline (BCP) were purchased from Xi'an Polymer Light Technology Corp (Xi'an, China). PCBM was purchased from Nano-C Inc. (Westwood, MA, USA), while poly(3,4-ethylenedioxythiophene) polystyrene sulfonate (PEDOT:PSS) (Clevious PVP AI 4083) was acquired from H.C. Starck company (Goslar, Germany). The Indium tin oxide (ITO) patterned glasses were bought from Ying Kou You Xuan Trade Co., Ltd. (Yingkou, China).

### 2.2. Device Fabrication

The ITO patterned glasses were cleaned with detergent, deionized water, and isopropanol (IPA), and each process was under the ultrasonic bath for 15 min, which was followed by the ultraviolet–ozone treatment for 20 min. PEDOT:PSS layer was spin-coated at 4000 rpm for 30 s followed by a thermal annealing treatment at 140 °C for 15 min. The precursor of $(BA)_2(MA)_3Pb_4I_{13}$ was prepared by dissolving BAI (0.5 M), $PbI_2$ (1.0 M), and MAI (0.75 M) in anhydrous mixed solvents (DMF: DMSO = 4:1, *v:v*), which were then stirred for 8 h at 70 °C followed by thermal annealing at 100 °C for 10 min. The perovskite film was formed by spin-coating the precursor solution at 1000 rpm for 6 s and then 6000 rpm for 60 s with the precursor in a nitrogen-filled glove box. During the spin coating, the PCBM solution (600 μL), which was prepared by dissolving PCBM (10, 20, and 30 mg mL$^{-1}$) in CB, was quickly dropped onto the sample at a delay time of 8 s. The samples were then thermally annealed on a hot plate at 100 °C for 20 min. Finally, Ag electrode with thickness of 100 nm was thermally coated on top of solar cells (working area of 0.1 cm$^2$) under high vacuum (<6.0 × 10$^{-6}$ Torr).

### 2.3. Characterization

Top-view scanning electron microscopy (SEM) images were measured with a JSM-7500F field-emission SE microscope (JOEL, Japan) at the acceleration voltage of 20 kV, while ultraviolet–visible (UV-vis) absorption spectra were checked with a UV-vis-near-infrared 3600 spectrometer (Shimadzu, Japan). The current density–voltage (*J-V*) characteristics of the 2D PSCs were recorded under an irradiation intensity of 100 mW cm$^{-2}$ (AM 1.5), while the electrochemical impedance spectroscopy (EIS) was measured using an electrochemical workstation (CH1622D Instruments, USA) under a bias voltage near the $V_{oc}$ of the devices in dark. The external quantum efficiency (EQE) of the PSCs was performed using a Solar Cell incident photon-to-current efficiency measurement system (Solar Cell Scan 100, Zolix, China).

### 3. Results and Discussion

To fabricate the inverted 2D PSCs, each functional layer was deposited in the stacking order of glass/ITO/PEDOT:PSS/perovskite/PCBM/BCP/Ag, as exhibited in Figure 1a. As shown in Figure 1b, the PSBM-assisted anti-solvent, which was prepared by dissolving PCBM powder into CB, was dropped on top of perovskite film during the spin-coating process to produce a perovskite/PCBM heterojunction. This approach integrated the two steps of casting anti-solvent and depositing ETL, which simplified the device fabrication process and yielded quality improved PCBM film.

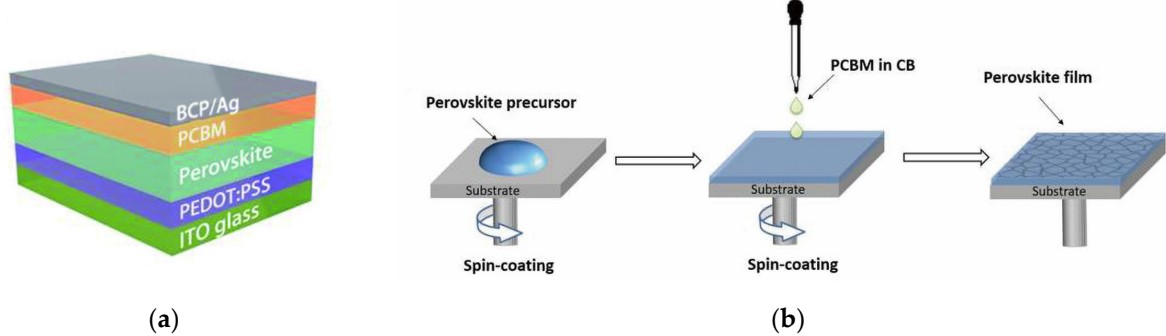

**Figure 1.** (**a**) Illustration of inverted-type 2D PSC with the configuration of glass/ITO/PEDOT:PSS/ perovskite/PCBM/BCP/Ag. (**b**) Schematic of perovskite deposition process using PCBM-assisted anti-solvent.

To estimate the influence caused by the one-step deposition method, J-V characteristics (reverse and forward scan) were measured. As shown in Figure 2a, applying the OSM resulted in an obviously improved photovoltaic performance, delivering a PCE of 12.8% along with $V_{oc}$ of 1.12 V, $J_{sc}$ of 15.69 mA cm$^{-2}$, and FF of 0.73 (reverse scan). As comparison, the PSC with PCBM deposited using regular method exhibited the PCE, $V_{oc}$, $J_{sc}$, and FF of 8.0%, 1.06 V, 11.45 mA cm$^{-2}$, and 0.66, respectively, in reverse scanning mode. The passivated perovskite film may contribute to the improved PCE. Note that the experimental group demonstrated a PCE of 11.9% in forward scanning mode, delivering the hysteresis index of 0.06 which can be calculated by the equation of $(PCE_{reverse} - PCE_{forward})/PCE_{reverse}$ [24]. However, the control group presented a higher hysteresis index of 0.09, suggesting that using the OSM is beneficial to suppress the hysteresis effect, which indicated the decreased defects. As exhibited in Figure 2c, the EQE spectra were recorded to compare the variation of photocurrents, in which the OSM based sample exhibited an obviously elevated photocurrent in almost the whole spectrum range compared with the control group. The integrated *Jsc* from EQE curves is 11.03 mA cm$^{-2}$ for generally formed PCBM and 15.15 mA cm$^{-2}$ for one-step formed PCBM, which agree with the data obtained in the J-V characteristics in Figure 2a. Furthermore, the stabilized photocurrent and PCE outputs of 2D PSCs using PCBM formed in different approaches were recorded. As shown in Figure 2c, the OSM based sample exhibited a stabilized photocurrent of 15.18 mA cm$^{-2}$ and PCE of 12.3%. This stabilized PCE is about 97% of its initial value obtained in J-V characteristics, which is slightly higher than that (96%) of the control group, corresponding with the suppressed hysteresis [21].

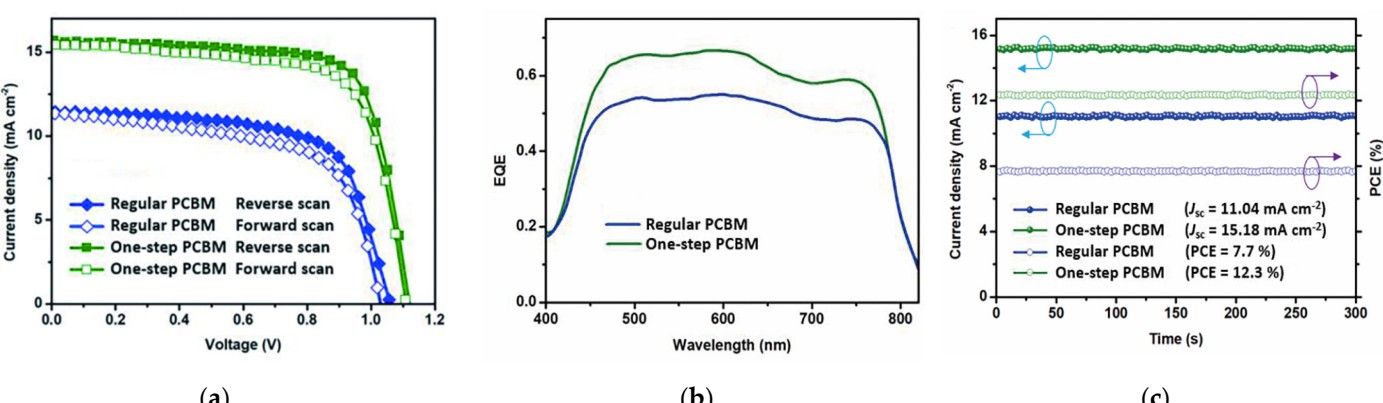

**Figure 2.** (**a**) J-V characteristics (reverse and forward scan), (**b**) EQE spectra, and (**c**) stabilized photocurrent as well as PCE outputs of 2D PSCs with PCBM formed with regular and one-step method.

To figure out the mechanism of the improved photovoltaic performance caused by the one-step deposited PCBM, the top-view SEM images were measured. As demonstrated in Figure 3a, the PCBM formed in regular method presented some gaps among the clusters, showing a poor film coverage, which may be caused by the robust hydrophobic character of 2D perovskite [27]. These gaps increased the opportunity of charge recombination and brought negative effects to device performance. As a contrast, the PCBM deposited with OSM exhibited a denser film with no gaps, which ensured a better electron transporting process. We deduce that the dense PCBM cluster is caused by the quickly dropped solution which is evaporated in a short time.

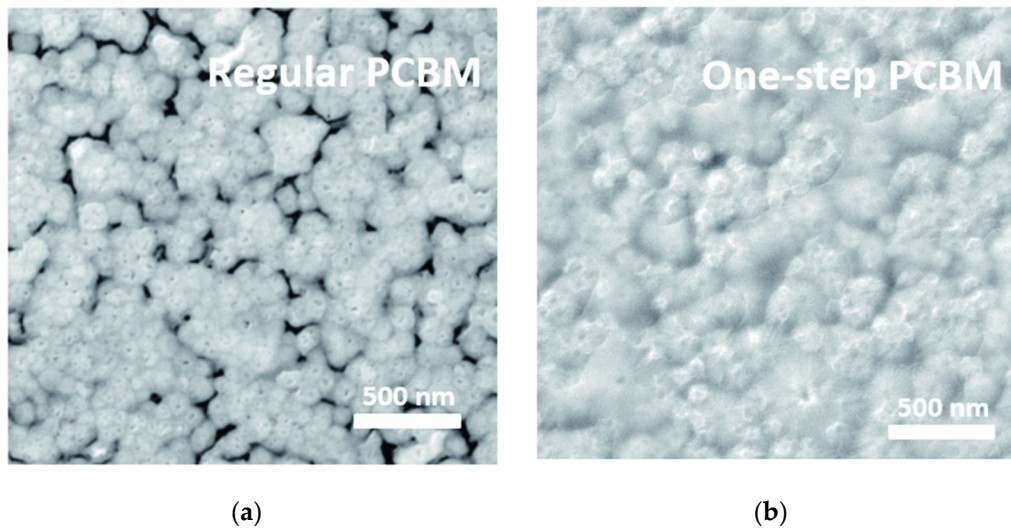

(**a**)                              (**b**)

**Figure 3.** Top-view SEM images for $(BA)_2(MA)_3Pb_4I_{13}$ deposited with (**a**) regular and (**b**) one-step method.

To further understand the variation about electrical performance affected by various film deposition methods, PL spectra were measured for the samples with configuration of perovskite, perovskite/PCBM (regular method), and perovskite/PCBM (one-step method). As shown in Figure 4a, all the emission peaks were around 740 nm which agreed with previous reports. An obviously shrinking effect occurred after depositing the PCBM (regular method) on top of perovskite film, suggesting the excellent electron transporting performance. When depositing PCBM with OSM, the emission peak decreased further, delivering the value around 70% of the PCBM sample (regular method). This decreased peak intensity may indicate that the OSM based PCBM exhibited a higher electron transporting performance [35,36]. In Figure 4b, the PCBM formed with regular method and OSM presented no obvious difference in UV absorbance. Note that the OSM based PCBM owned a slightly higher absorbance than the regular PCBM around the region of 2D perovskite characteristic peak (around 600nm) which can be observed from the inset of Figure 4b. It is deduced that the slightly increased absorbance may arise from a better film coverage of OSM based PCBM [37]. This result agreed with the SEM data in Figure 3, i.e., the OSM based PCBM was more compact than the generally formed PCBM. We also analyzed the EIS to study the variation of electric performance. As shown in Figure S1, the Nyquist plots with two characteristic arcs and equivalent circuit were exhibited, from which the recombination resistance ($R_{rec}$), series resistance ($R_s$), and capacitor C can be observed. The OSM based sample exhibited a $R_s$ of 27.4 Ω and $R_{ct}$ of 1604.7 Ω, while the regular sample presented a $R_s$ of 31.6 Ω and $R_{ct}$ of 1158.5 Ω. The decreased $R_s$ and increased $R_{rec}$ indicated the improved electron transporting performance and suppressed charge recombination, respectively, probably suggesting the defect-passivated perovskite [38].

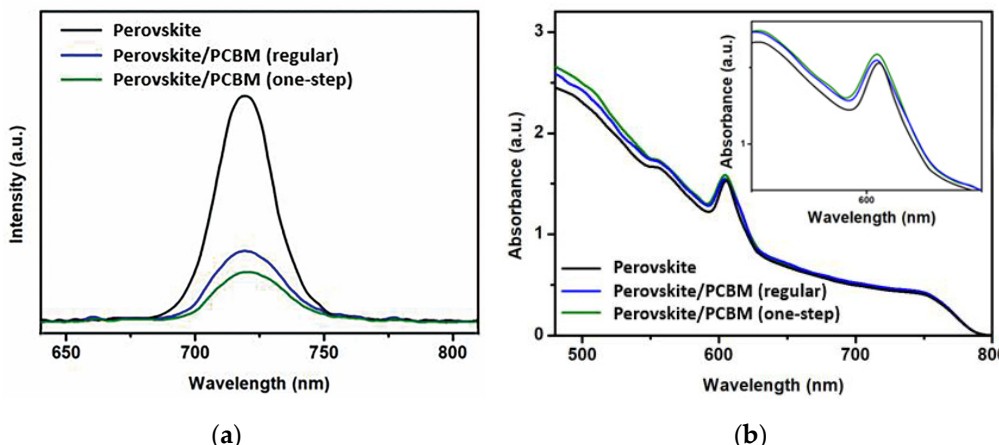

(a)

(b)

**Figure 4.** (**a**) PL spectra and (**b**) UV absorbance of samples with configuration of perovskite, perovskite/PCBM (regular method), and perovskite/PCBM (one-step method).

To search for the optimized PCE, we measured the J–V characteristics for OSM formed PCBM using different concentrations. To obtain the more obvious changes in data, we used 10 mg mL$^{-1}$ increment of the concentrations. As shown in Figure 5a, when the concentrations of PCBM solution were increased from 10 to 30 mg mL$^{-1}$, the PCEs were elevated from 10.2% to 12.8% and then rolled back to 10.5%. The excess solution concentration may lead to unsuitable film thickness, hindering the effective electron transportation.

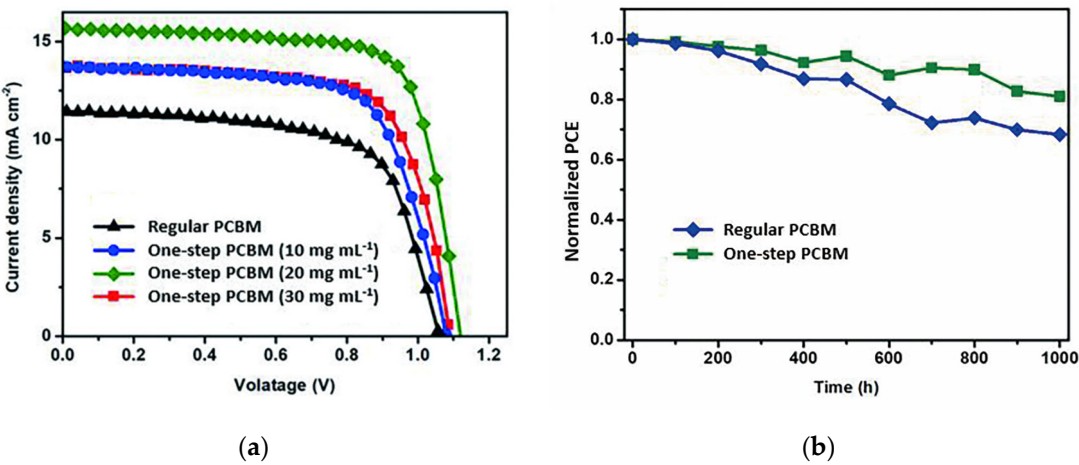

(a)

(b)

**Figure 5.** (**a**) J-V characteristics of 2D devices based on ETLs deposited under different parameters. (**b**) Long-term stability test for perovskite devices based on different PCBM deposition methods for 1000 h under illumination at around 25 °C without encapsulation (about 30% humidity).

In addition to improved PCE, the environmental stability of devices is also essential for the device performance. Therefore, we evaluated the long-term stability of perovskite devices based on different PCBM deposition methods for 1000 h under illumination at around 25 °C without encapsulation (about 30% humidity). As shown in Figure 5b, the experimental group exhibited a slower degradation rate, retaining about 81% of the initial PCE, while the control group maintained only about 68% of the initial PCE. This can be explained by the fact that the OSM based sample owned the defect-suppressed perovskite and quality-improved PCBM, suppressing the degradation that occurred from the defects.

## 4. Conclusions

In summary, we reported a facile method to form the perovskite/PCBM heterojunction with better quality using the OSM. The defects of perovskite film were passivated and the morphology of PCBM was improved with higher film coverage and less gaps, ensuring

a better electron transporting performance and suppressed charge recombination. As a result, the OSM based 2D PSC delivered a PCE of 12.8% along with $V_{oc}$ of 1.12 V, $J_{sc}$ of 15.69 mA cm$^{-2}$, and FF of 0.73 (reverse scan). Thanks to the defect-passivated perovskite and quality-improved PCBM, the degradation rate was effectively suppressed. After storing the samples for 1000 h under illumination at around 25 °C without encapsulation, the experimental group retained about 81% of the initial PCE, which is higher than that (68%) of control group.

**Supplementary Materials:** The following are available online at https://www.mdpi.com/article/10.3390/cryst11080933/s1, Figure S1: Nyquist plots of ETL fabricated in different approaches in 2D PSCs.

**Author Contributions:** Conceptualization, F.Z. and X.X.; methodology, C.X.; investigation, H.W.; data curation, G.L.; writing—original draft preparation, H.W.; writing—review and editing, S.W.; supervision, S.W.; project administration, F.Z.; funding acquisition, F.Z. and X.X. All authors have read and agreed to the published version of the manuscript.

**Funding:** This work was supported by the National Research Foundation of Korea (NRF) funded by the Ministry of Science and ICT (Grant No.: 2020R1GA1102816), and the Department of Science & Technology of Jilin Province (Grant No.: 20160414043GH).

**Institutional Review Board Statement:** Not applicable.

**Informed Consent Statement:** Not applicable.

**Data Availability Statement:** Data are available on request from the authors.

**Acknowledgments:** We acknowledge the financial support from the National Research Foundation of Korea (NRF).

**Conflicts of Interest:** The authors declare no conflict of interest. We identify and declare that no personal circumstances or interest that may be perceived as inappropriately influencing the representation or interpretation of reported research results. Additionally, the funders had no role in the design of the study; in the collection, analyses, or interpretation of data; in the writing of the manuscript, or in the decision to publish the results.

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
