# Peer review of "Surface Passivation Using N-Type Organic Semiconductor by One-Step Method in Two-Dimensional Perovskite Solar Cells"

_crystals, doi:10.3390/cryst11080933_

Round 1

Reviewer 1 Report

This is an excellent work. The methods, results and discussions are very good.

My only technical concern is on the optimization experiment. The 10 mg/mL increment of the concentrations should have been smaller. I am just wondering if an optimum power conversion efficiency (PCE) could have been obtained at 22 mg/mL or 25 mg/mL or at other concentration near 20 mg/mL. You may put this into consideration in future works.

I have a couple comments on typo errors:

In Line 4, Xie,3,* and* should be Xie,3,** and

All references at the end of a sentence should come before the full stop "." This should be corrected throughout the paper. As an example in Line 39 practical application.[8-12] However should be written as practical application [8-12]. However

Author Response

Response: Thank you for your valuable comment. We admit that a smaller increment is better to find the optimized PCE. In fact, we have tried some other concentrations, such as 5 mg/mL, 15 mg/mL, and 25 mg/mL in the previous experiments. However, the PCE and other electrical results obtained from these concentrations didn’t exhibit clear difference compared with their neighbor groups. Hence, we used the 10 mg/mL increment of the concentrations to make the charts clearer. To clarify this issue, we added one sentence such as “To obtain the more obvious changes in data, we used 10 mg mL-1 increment of the concentrations” (L 208-209)

We modified the typo errors such as “Xiaoyin Xie,3,** and Sheng Wu,4,***”, and moved the all the citations into the sentences.

Reviewer 2 Report

I think it's good that the one-step method has improved various properties. However, the result is supported only by the PL and absorption spectrum in Fig. 4, which I think is insufficient.

Regarding the part that the decrease in the intensity of the PL spectrum is related to the high electron transport performance, please support the conclusion by showing references.

L188-191Probskite / PCBM produced by the one-step method is claimed to show slightly higher absorption intensity with the SEM image in Figure 3.  It is unclear why this slight change in absorption spectrum leads to the conclusion that OSM-based PCBM is more compact than commonly formed PCBM.  Please support your conclusion by adding a description and providing a reference.

If authors discuss the change in electronic characteristics only by the absorption spectrum, it is necessary to expand the relevant area in Fig. 4 (b), which the author is paying attention to, and discuss changes in line width and shape, for example, to clarify the relationship with Fig. 3. 

Author Response

  • I think it's good that the one-step method has improved various properties. However, the result is supported only by the PL and absorption spectrum in Fig. 4, which I think is insufficient. Regarding the part that the decrease in the intensity of the PL spectrum is related to the high electron transport performance, please support the conclusion by showing references.

Response: Thank you for your valuable comment. It is true that PL result and related analysis to the decrease in the intensity of the PL spectrum is insufficient. To clarify the mechanism, we added more sentences and citations such as “This decreased peak intensity may indicate that the OSM based PCBM exhibited a higher electron transporting performance [35, 36].” (L189-190)

  • L188-191Probskite / PCBM produced by the one-step method is claimed to show slightly higher absorption intensity with the SEM image in Figure 3.  It is unclear why this slight change in absorption spectrum leads to the conclusion that OSM-based PCBM is more compact than commonly formed PCBM.  Please support your conclusion by adding a description and providing a reference.

Response: Thank you for your valuable comment. We admit that it is needed to add more description and provide a reference to clarify this phenomenon. Based on this comment, we added one sentence provided a reference such as “It is deduced that the slightly increased absorbance may arise from a better film coverage of OSM based PCBM [37].” (L 194-195)

  • If authors discuss the change in electronic characteristics only by the absorption spectrum, it is necessary to expand the relevant area in Fig. 4 (b), which the author is paying attention to, and discuss changes in line width and shape, for example, to clarify the relationship with Fig. 3.

Response: Thank you for your valuable comment. We admit that is necessary to expand the relevant area in Fig. 4 (b) and provide more discussion. Based this comment, we expanded the relevant area in Fig. 4 (b) and provided more discussion such as “In Fig. 4(b), the PCBM formed with regular method and OSM presented no obvious difference in UV absorbance. Note that the OSM based PCBM owned a slightly higher absorbance than the regular PCBM around the region of 2D perovskite characteristic peak (around 600nm) which can be observed from the inset of Fig. 4(b).” (L 190-194)

Reviewer 3 Report

 Zhou et al previously reported a unique solvent engineering method to form perovskite/fullerene heterojunction in one step using PCBM ( DOI: 10.1039/C7NR07753J) and more importantly, the detailed mechanism was deeply revealed.

This work only provided a similar method to prepare the perovskite film and experimental characterizations used to reveal the effects of this method are also missing.

There is no novelty in this work. So I think this work is not suitable for publication in the current state.

Author Response

Response: Thank you for your valuable comment. We carefully studied Zhou’s thesis (DOI:10.1039/C7NR07753J) and admit that it is a good work. It is true that we all studied a unique solvent engineering method by forming the perovskite/fullerene heterojunction in one step. However, there are a lot of different points and the research focuses of these two theses are different. Firstly, we studied the effect of OSM based PCBM on the 2D perovskite which is the different research subject from the previous work and owns quite different physical and chemical properties from the 3D perovskite. The 2D perovskite is usually more hydrophobic than 3D perovskite, which makes the situation and fabrication difficulty different for depositing a film on top of the 2D perovskite. Secondly, we studied more about the PCBM morphology formed with different methods and discussed its relationship with other electrochemical data. The previous work studied more about the effects of different methods on perovskite quality. Finally, we modified some errors and added more discussions to make this manuscript more readable. In general, we deem that our work may provide some extensions compared with the previous work and owns the necessary novelty for publication.

Round 2

Reviewer 2 Report

The manuscript is now ready for publication.

Reviewer 3 Report

I may accept this argument, however, in the revised manuscript, the authors did not still add the reported work(DOI:10.1039/C7NR07753) as a reference and give some discussions. Thus I think this is very important to give enough affirmation for the reported work and provide useful information for the readers. If this discussion is added to the manuscript, I think this manuscript can be published.